# Survival and predictors of breast cancer mortality in South Ethiopia: A retrospective cohort study

Abel Shita[1,2,3], Alemayehu Worku Yalew[2], Edom Seife[4], Tsion Afework[2,3,5],
Aragaw Tesfaw[6], Zenawi Hagos Gufue[7], Friedemann Rabe[3,8], Lesley Taylor[9],
Eva Johanna Kantelhardt[3,8,10], Sefonias Getachew[2,3,5,8]*

1 Mizan Aman College of Health Sciences, Department of Public Health, Addis Ababa, Southwest Ethiopia,
2 Department of Preventive Medicine, School of Public Health, Addis Ababa University, Addis Ababa,
Ethiopia, 3 Global Health Working Group, Martin-Luther-University, Halle (Saale), Germany, 4 Department
of Medicine, Oncology Center, Addis Ababa University, Addis Ababa, Ethiopia, 5 NCD Working Group
School of Public Health Addis Ababa University, Addis Ababa, Ethiopia, 6 Department of Public Health,
College of Health Science, Debre Tabor University, Debra Tabor, North West Ethiopia, 7 Department of
Public Health, College of Medicine and Health Sciences, Adigrat University, Adigrat, Ethiopia, 8 Institute of
Medical Epidemiology, Biometrics and Informatics, Martin Luther University, Halle, Germany, 9 City of Hope
National Medical Center, Duarte, Los Angeles County, California, United States of America, 10 Department
of Gynaecology, Martin Luther University, Halle, Germany

* safoget@yahoo.com

pone.0282746

Communicable Diseases Research Center,
Endocrinology and Metabolism Research Institute,
Tehran University of Medical Sciences, ISLAMIC
REPUBLIC OF IRAN

## Abstract

### Background

Breast cancer is the most frequently diagnosed cancer and the leading cause of cancer
death in over 100 countries. In March 2021, the World Health Organization called on the
global community to decrease mortality by 2.5% per year. Despite the high burden of the
disease, the survival status and the predictors for mortality are not yet fully determined in
many countries in Sub-Saharan Africa, including Ethiopia. Here, we report the survival sta-
tus and predictors of mortality among breast cancer patients in South Ethiopia as crucial
baseline data to be used for the design and monitoring of interventions to improve early
detection, diagnosis, and treatment capacity.

### Methods

A hospital-based retrospective cohort study was conducted among 302 female breast can-
cer patients diagnosed from 2013 to 2018 by reviewing their medical records and telephone
interviews. The median survival time was estimated using the Kaplan-Meier survival analy-
sis method. A log-rank test was used to compare the observed differences in survival time
among different groups. The Cox proportional hazards regression model was used to iden-
tify predictors of mortality. Results are presented using the crude and adjusted as hazard
ratios along with their corresponding 95% confidence intervals. Sensitivity analysis was per-
formed with the assumption that loss to follow-up patients might die 3 months after the last
hospital visit.

**Data Availability Statement:** The data underlying the results presented in the study will be available from the primary or corresponding author upon reasonable request. The study was focused on two hospitals in South part of the country and it has the potential of identifying the anonymized information of patients which is not granted. Data are available also from the Ethics Committee of the School of public health Addis Ababa University (publichealth@aau.edu.et) for researchers who meet the criteria for access to confidential data.

**Funding:** The study was funded by the School of Public Health, Addis Ababa University, Ethiopia as part of a graduate studies program. The stipend for Sefonias Getachew as senior author was also supported by the Else-Kroener Foundation through Martin Luther University, Halle-Wittenberg, Germany, grant No. 2018_HA31SP. The funders had no role in study design, data collection, and analysis, decision to publish, or preparation of the manuscript.

**Competing interests:** The authors have declared that no competing interests exist.

## Results

The study participants were followed for a total of 4,685.62 person-months. The median survival time was 50.81 months, which declined to 30.57 months in the worst-case analysis. About 83.4% of patients had advanced-stage disease at presentation. The overall survival probability of patients at two and three years was 73.2% and 63.0% respectively. Independent predictors of mortality were: patients residing in rural areas (adjusted hazard ratio = 2.71, 95% CI: 1.44, 5.09), travel time to a health facility ≥7 hours (adjusted hazard ratio = 3.42, 95% CI: 1.05, 11.10), those who presented within 7–23 months after the onset of symptoms (adjusted hazard ratio = 2.63, 95% CI: 1.22, 5.64), those who presented more than 23 months after the onset of symptoms (adjusted hazard ratio = 2.37, 95% CI: 1.00, 5.59), advanced stage at presentation (adjusted hazard ratio = 3.01, 95% CI: 1.05, 8.59), and patients who never received chemotherapy (adjusted hazard ratio = 6.69, 95% CI: 2.20, 20.30).

## Conclusion

Beyond three years after diagnosis, patients from southern Ethiopia had a survival rate of less than 60% despite treatment at a tertiary health facility. It is imperative to improve the early detection, diagnosis, and treatment capacities for breast cancer patients to prevent premature death in these women.

## Background

Breast cancer (BC) is currently the most common malignancy worldwide, with the majority of premature deaths occurring in Sub-Saharan Africa [1]. A recent prospective study across the region estimated that one-third of premature deaths in the region could be averted over the next decade with early diagnosis and improved access to treatment [2]. In contrast to low-income countries, high-income countries have achieved a 40% decrease in breast cancer mortality since the 1980s in regions where 60% of cases were diagnosed at stage I or II and public health care systems provided adequate treatment [3].

Given these data, the World Health Organization (WHO) Global Breast Cancer Initiative was launched in 2021, assembling global collaborations to decrease mortality by 2.3% per year. To achieve this goal in Ethiopia, comprehensive baseline data on survival and predictors for mortality is urgently needed for designing and monitoring interventions. Ethiopia has strengthened its capacity for the treatment of breast cancer patients over the last decade in conditions of extremely limited resources [4]. It is the second most populous country in Sub-Saharan Africa, with one of the lowest gross domestic products (GDP) and lowest ratios of healthcare providers per person [5].

For a population of over 114 million, there are currently 23 medical oncologists trained to provide both chemotherapy and radiation. Nearly 80% of the population resides in rural areas. In 2016, Ethiopia developed a National Cancer Control Plan to increase early detection and expand treatment capacity to regional referral centres around the country. Since then, access to chemotherapy and endocrine medications has improved, such as adriamycin, cyclophosphamide, taxanes, aromatase inhibitors, and tamoxifen. The capacity for manual immunohistochemical staining to determine breast cancer phenotype is growing but extremely limited.

Despite these improvements, many barriers to breast cancer treatment exist–these include delays in diagnosis, disjointed patient navigation, financial toxicity, the stigma of treatments, and lack of treatment completion or adherence [6].

The average patient is young, i.e. 30–45 years old. There are an average of 16,000 new breast cancer cases per year, though the actual national burden of the disease may be closer to 30,000 [7]. Most patients present with stage III or IV disease [8]. Scholars are warning that unless urgent action is taken, BC will compound Sub-Saharan Africa's disease burden, increase poverty and gender inequality, as well as reverse the current global gains against maternal and neonatal mortality [9]. Importantly, 2/3 of breast cancer cases in Ethiopia have a hormone receptor positive phenotype; these are potentially curable if treated early with mastectomy and endocrine therapy costing $7 per day [10].

To our knowledge, only a few studies exist on the survival of breast cancer patients in Ethiopia. In a 2014 report, rural patients with access to surgery alone had a two-year survival rate of 46% compared to women treated in Addis Ababa, who had a survival rate of 74% after treatment with chemotherapy, surgery, and endocrine therapy [10]. At that time, the preferred chemotherapy regimen was FAC (5-fluorouracil, anthracycline, cyclophosphamide), but AC and CMF were also given (anthracycline with cyclophosphamide or cyclophosphamide, methotrexate, and 5-fluorouracil). Surveillance of cancer survival is critical for developing and monitoring interventions to improve early detection and treatment capacity. In another study in northwest Ethiopia, the overall survival at two years was 54% [11].

Survival data are used to formulate cancer control strategies, prioritize cancer control measures, and assess both the effectiveness and cost-effectiveness of those strategies [12]. This study will fill information gaps by estimating overall survival and identifying predictors of mortality in settings outside of Addis Ababa that have surgery and contemporary systemic endocrine and chemotherapy, notably taxanes. Moreover, it will be used as a baseline for future evaluation of the quality of care and the progressive development of healthcare systems in the country. Researching interventions to improve survival in Ethiopia not only strengthens breast cancer care and education in the country, but also contributes to regional knowledge about effective strategies to address the problem in Sub-Saharan Africa. Therefore, this study aimed to determine the survival and predictors of mortality among breast cancer patients in southern Ethiopia.

## Methods

### Study area and approach

A hospital-based retrospective cohort study was conducted at the oncology units of Hawassa University Comprehensive Specialized Hospital (HUCSH) and Yanet Internal Medicine Specialized Center (YIMSC) which are located in Hawassa city, the former SNNP Regional State of Ethiopia, and the recent Sidama regional state. Based on the 2007 Census conducted by the Central Statistical Agency of Ethiopia (CSA), the SNNPR has an estimated total population of 14,929,548 [13]. Hawassa city is located 273 kilometres south of the capital city, Addis Ababa. Both health facilities provide diagnostic services as well as surgical and chemotherapy treatment for cancer patients. They are referral centres for all cancer cases in the southern part of the country. For radiotherapy, patients who can afford the service were referred to Tikur Anbesa Specialized Hospital, Addis Ababa. The data extraction period was from February 8th to April 30th, 2019.

### Sampling and data collection procedures

There were 337 patients with BC diagnosed from January 1, 2013, to December 30, 2018, at HUCSH and YIMSC. After the exclusion of incomplete charts and male BC patients, 302

female breast cancer patient charts were reviewed (Fig 1). To ascertain the outcome, a telephone interview was performed with 206 patients or with their relatives who were >18 years of age. The remaining 96 patients were unavailable by telephone. The second review of patient charts was performed 3 months after the end of the chart review to see whether any of the remaining 96 patients had subsequently visited the hospital. Of these 96, we identified 39 patients who had visited the hospital, thereby confirming they were alive.

### Variables and operational definitions

The outcome variable was time to death. Explanatory variables included four categories: 1) socio-demographic data (age, place of residence, marital status, level of education, religion, travel time to hospital, and occupation); 2) clinical and pathological characteristics of the disease (duration of symptom, stage of BC at diagnosis, tumour size, histological type, nodal status, nuclear grade, and distant metastasis); 3) type of treatment (adherence to chemotherapy, surgical therapy, and hormonal therapy); 4) co-morbidities (human immunodeficiency virus (HIV), hypertension, diabetes mellitus, asthma).

Survival time was defined as the total time the patient lived after diagnosis. Follow-up time was defined as the interval of time from entry in the medical record to the date of death or the end of the study. An event was defined as patient death regardless of cause. In the worst-case scenario, patients lost to follow-up (LTF) that were unavailable by telephone and did not visit the hospital 3 months after the conclusion of our chart review were considered deceased [10, 14].

We defined patients as lost to follow-up (LTF) who could not be located for >6 months. Good adherence to chemotherapy was defined when patients completed all cycles of chemotherapy per the standard of care guidelines and poor adherence to chemotherapy was defined when patients did not complete all cycles of chemotherapy [15]. Similar to prior work, we performed models of best-case and worst-case scenarios to estimate survival. Data were categorized as right or left censored. Right-censored data included patients alive at the end of the study. In

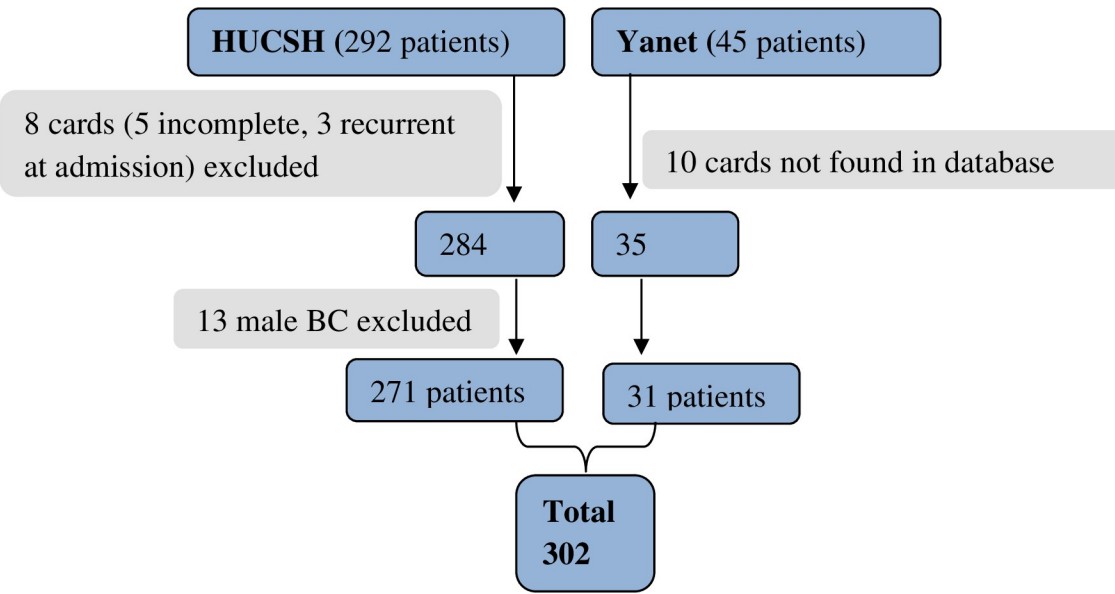

**Fig 1. Schematic presentation of the sampling procedure used to determine the survival and predictors of mortality among female breast cancer patients in Southern Ethiopia, 2019.**

the best-case analysis, patients LTF was right-censored and considered alive. In the worst-case analysis, we assumed patients LTF died 3 months after their last hospital visit [10, 16].

## Measurements

The stage of BC was determined by the American Joint Committee on Cancer staging system AJCC (seventh edition) using the information on tumour size (T) and nodal status (N) and metastasis (M) [17]. Two observations were used in this regard: TNM staging at the time of diagnosis and the last follow-up to see if there was a progression of the disease. Tumour size was primarily ascertained by clinical examination of the oncologist. If that documentation was not available, the information was obtained from surgical pathology reports. Data on histological type and nuclear grading were also obtained from surgical pathology reports.

Patients with non-metastatic BC were treated with eight cycles of adjuvant chemotherapy consisting of four cycles of AC (adriamycin + cyclophosphamide) and four cycles of taxol. Those with metastatic BC are treated with six cycles of AC. The current status of patients (alive or deceased) was confirmed by telephone. For those not available by telephone, the survival status was collected from the medical chart by checking whether the patient had a follow-up visit 3 months after the end date of the study period; if this was the case, the patient was considered alive.

## Data collection tools and procedure

A structured record review checklist was developed after a review of the literature and by assessing the availability of information from medical records. For the telephone interview, a questionnaire was prepared based on the required information in English and translated by experts to local languages (Amharic, Sidama, and Affan Oromo) and then back to English to maintain consistency. All medical records of patients with BC were identified by their medical record number, reviewed for eligibility, and the information was extracted. A phone call was made to patients with BC or their families to assure their current living status and to conduct an interview.

## Data analysis

The collected data were coded and checked for clarity, consistency, and completeness up to the end of each data collection period and Epi info Version 7.2.2.6 was used for data entry. The entered data was exported to STATA version 14.0 for windows. Descriptive statistics of numeric variables were presented in medians with interquartile range (IQR), and categorical variables were presented using frequency and percentages.

The overall survival was estimated with Kaplan-Meier survival curves. A log-rank test was used to compare the survival time among groups with a 95% CI. The assumptions of the Cox proportional hazards regression model were checked by log (-log (St) plots, and the Schoenfeld residual test. Covariates that did not violate the assumption test and had a P-value <0.25 significant level on bivariate Cox regression were considered for multivariable analysis. A P-value <0.05 was considered to denote statistical significance. Worst-case and best-case scenarios were modelled to calculate survival rates. Multi-collinearity and interactions for the main effect model were checked, and the variance inflation factor greater than 10 was considered to denote its existence. We used the Cox-Snell residual plot to check the goodness of fit of the final model.

## Ethical approval

Ethical approval was secured from the research and ethical committee of Addis Ababa University, School of Public Health, with approval number 0054/2019. After an in-depth explanation

of the aim of the study, formal permission was obtained from HUCSH and YIMSC to review patient records and contact patients via telephone calls. Verbal consent was obtained via the telephone from patients or, for patients who had died, from the patient's relatives (father, mother, husband, or children > 18 years old). All information was kept confidential.

## Results

### Patient characteristics

Out of 302 female breast cancer patients included in the study, a telephone interview was conducted with 206 patients or their relatives. The second review of patient charts was made three months after the end of the first chart review. We found that 39 (12.9%) patients who were unavailable by telephone call had subsequently visited the hospital. Accordingly, the outcome of 245 (81.1%) patients was confirmed: 178 (58.9%) were alive, 67 (22.2%) were deceased, and 57 (18.9%) patients were LTF.

### Socio-demographic and clinical characteristics

Out of 302 women with BC recruited for this study, the majority 177 (58.6%) were within the age of 35–50 years, the median age being 39 years (IQR = 32–45). We found that 239 (79.1%) of patients were pre-menopausal. Ninety-one (30.1%) patients were diagnosed between 2013and 2015 whereas 211 (69.9%) were diagnosed between 2016 and 2018. Of the 302 BC patients, 252 (83.4%) presented with advanced stage at the first hospital visit: 161(53.3%) with stage III and 91(30.1%) with stage IV disease. Thirty-five (11.6%) patients were diagnosed with comorbidity, out of which 23 (7.6%) had hypertension and 5 (1.6%) had HIV.

Not all socio-demographic information and clinical characteristics were available from the medical records. The available information on residence from 263 patients indicated that 189 (71.9%) were from urban areas. Their median travel time to the Hawassa referral centre for cancer treatment was 2 hours (IQR = 0.30–3.30 hours). Data on marital status was available for 208 patients; of these, 178 (85.6%) were married, 108 (51.9%) were housewives, and 78 (37.5%) could neither read nor write.

Data on symptoms was available for 283 patients; out of these, 208 (73.5%) patients presented with complaints of a breast lump. Data on the duration of symptoms before seeking diagnosis was available for 283 patients; out of these, 112 (39.6%) sought care at the oncology unit within 7–23 months of symptom onset.

Out of 196 patients whose tumour grade was available, 109 (55.6%) were grade II (moderately differentiated). Data on tumour size was available for 294 patients; of these, 173 (58.8%) had a tumour size >5 cm. Nodal status was documented for a total of 285 patients, of whom 240 (84.2%) had a nodal disease. A total of 297 records contained a morphologic classification of BC, of which 249 (83.8%) were documented as ductal carcinoma, 24 (8.1%) as lobular carcinoma, and 24 (8.1%) as mixed ductal and lobular (Table 1).

### Diagnosis and treatment of patients

A total of 208 patients with BC were treated with surgery. Out of the 173 (83.2%) who underwent modified radical mastectomy (MRM), 17 (8.2%) had a 'toilet' mastectomy and the rest 18 (8.6%) had no detailed information on surgical therapy. Data on chemotherapy was available for 275 patients. Chemotherapy was administered to 219 (79.6%) patients, out of whom 138 (63%) discontinued treatment, 54 (24.7%) completed the treatment, and 27 (12.3%) were in active treatment during data collection. Data on hormonal therapy was available for 302 patients and indicated that 202 (66.9%) did not receive treatment. Out of 100 patients that

**Table 1. Baseline characteristics of female breast cancer patients in southern Ethiopia, 2013–2018 (n = 302).**

| Baseline charactersistics | Frequency | Percentage |
|---|---|---|
| Age (in years) (n = 302) | | |
| <35 | 82 | 27.2 |
| 35–50 | 177 | 58.6 |
| >50 | 43 | 14.2 |
| Residence (n = 263) | | |
| Urban | 189 | 71.9 |
| Rural | 74 | 28.1 |
| Travel time (n = 263) | | |
| < 3 hours | 197 | 74.9 |
| 3–6 hours | 51 | 19.4 |
| ≥ 7 hours | 15 | 5.7 |
| Menopausal status | | |
| Pre-menopause | 239 | 79.1 |
| Post-menopause | 63 | 20.9 |
| Breast complaint at the first visit (n = 283) | | |
| Breast lump | 208 | 73.5 |
| Breast ulcer | 34 | 12.0 |
| Others * | 41 | 14.5 |
| Duration of symptoms (n = 283) | | |
| 0–6 months | 110 | 38.9 |
| 7–23 months | 112 | 39.6 |
| > 23 months | 61 | 21.5 |
| Histological type (n = 297) | | |
| Ductal | 249 | 83.8 |
| Lobular | 24 | 8.1 |
| Others** | 24 | 8.1 |
| Stage of breast cancer at diagnosis (n = 302) | | |
| Early | 50 | 16.6 |
| Advanced | 252 | 83.4 |
| Tumour size (n = 294) | | |
| TI/II | 121 | 41.2 |
| TIII/IV | 173 | 58.8 |
| Distant metastasis (n = 302) | | |
| Yes | 56 | 18.5 |
| No | 246 | 81.5 |
| Nodal status (n = 285) | | |
| Positive | 240 | 84.2 |
| Negative | 45 | 15.8 |
| Co-morbidities (n = 302) | | |
| Yes | 35 | 11.6 |
| No | 267 | 88.4 |

* = shortness of breath, axillary swelling, nipple retraction, nipple discharge.

** = Mixed ductal & lobular, mucinous

took hormonal therapy, 65(65%) were treated with tamoxifen and 35% (n = 35) with anastrozole. Eight patients received radiation therapy at Tikur Anbesa Specialized Hospital (Table 2).

**Table 2. Treatment characteristics of female breast cancer patients in southern Ethiopia, 2013–2018 (n = 302).**

| Charactersistics | Frequency | Percent |
|---|---|---|
| Surgical therapy | | |
| Yes | 208 | 68.9 |
| No | 94 | 31.1 |
| Type of surgery | | |
| MRM | 173 | 83.2 |
| Toilet mastectomy | 17 | 8.2 |
| Document not found | 18 | 8.6 |
| Chemotherapy | | |
| Yes | 192 | 69.8 |
| No | 83 | 30.2 |
| Hormonal therapy | | |
| Yes | 100 | 33.1 |
| No | 202 | 66.9 |
| Hormonal therapy type | | |
| Tamoxifen | 65 | 21.5 |
| Anastrazole | 35 | 11.6 |
| Not taking | 202 | 66.9 |
| Radiation | | |
| Yes | 8 | 2.7 |
| No | 294 | 97.3 |
| Adherence to chemotherapy | | |
| Good adherence | 54 | 28.1 |
| Poor adherence | 138 | 71.9 |
| Recurrence | | |
| Yes | 22 | 7.3 |
| No | 280 | 92.7 |
| Progression | | |
| Yes | 40 | 13.3 |
| No | 262 | 86.7 |

Abbreviations: MRM: modified radical mastectomy

Of the total of 302 patients, progression of disease was documented for 40 patients (13.2%) and recurrence for 22 (7.3%).

## Clinical profiles of patients lost to follow-up (LFT)

A total of 57 patients were lost to follow-up for >6 months. Of these, 55 (96.5%) had advanced-stage disease at diagnosis. The majority 68.4% (n = 39) received breast surgery. Only 15.8% (n = 9) of patients received hormonal therapy.

## Survival status

In the best-case scenario, a total of 302 BC patients were observed for 4685.61 person-months or at-risk time. Over a total follow-up time of 5 years, 67 patients died. The median follow-up time was 50.81 months (IQR = 18.38—not estimable). The overall survival of BC patients at the end of 1 year was 83.0%; at 2 years 73.2%; at 3 years 63.1%. The overall two-year survival of patients with early-stage BC was 89.9%, and the advanced-stage was 63.8%. Specifically for patients with Stage III disease, it was 73.4% and for Stage IV it was 44.3% (Table 3).

In the worst-case scenario, a total of 124 patients were considered deceased. In this analysis, the median survival of BC patients was 30.57 months (IQR = 7.23–64.23). The overall survival at the end of 1 year was 67%; at 2 years 51.3%; at 3 years 44.6% (Fig 2). The overall two-year survival for the early-stage BC was 85.1%; and the advanced-stage 44.1%. Specifically for patients with Stage III disease, it was 55.1%, and for Stage IV 23.7%. (Fig 3).

## Predictors of survival

The hazard ratio of death of rural patients was 2.7 times higher than urban dwellers after adjusting for age, histological type, surgery, and nodal status (adjusted hazard ratio (AHR) = 2.71, 9% CI: 1.44, 5.09). Patients who travelled >7 hours to Hawassa for cancer treatment had 3.42 times increased risk of death compared to those who travelled less than 3 hours (AHR = 3.42, 95% CI; 1.05, 11.10). Patients who presented to the oncology unit within 6 months of symptom onset had better survival rates: those presenting at 7 to 23 months had a 2.63 times increased risk of death (AHR = 2.63, 95% CI; 1.22, 5.64), and those presenting after 23 months had a 2.37 times increased risk of death (AHR = 2.37, 95% CI; 1.00, 5.59). Patients with advanced-stage disease had a 3.01 times higher risk of death compared to those with early-stage disease (AHR = 3.01, 95% CI; 1.05, 8.59). The hazard of death for patients who received no chemotherapy was 6.69 times (AHR = 6.69, 95% CI; 2.20, 20.30) higher than those patients who had good adherence to chemotherapy (Table 4).

**Table 3. Log-rank test for equality of survival function of female breast cancer patients in southern Ethiopia, 2013–2018 (n = 302).**

| Covariates | Survival time | | Observed death | Expected death | P-value |
|---|---|---|---|---|---|
| | 2 years | 3 years | | | |
| Overall survival | 73.2 (65.8, 79.3) | 63.0 (53.3, 71.2) | | | |
| Residence | | | | | |
| Urban | 80.1 (71.6, 86.3) | 71.4 (80.0, 60.1) | 36 | 52.13 | <0.001 |
| Rural | 50.5 (34.2, 64.7) | 27.5 (07.4, 52.6) | 30 | 13.87 | |
| Age in years | | | | | |
| < 35 | 61.5 (43.9, 75.0) | 56.7 (38.2, 71.6) | 23 | 14.06 | 0.019 |
| 35–50 | 76.0(66.5, 83.1) | 70.2 (58.9, 78.9) | 34 | 43.65 | |
| > 50 | 81.5 (63.0, 91.3) | 48.8 (21.1, 71.8) | 10 | 9.28 | |
| Stage at diagnosis | | | | | |
| Early | 95.1 (81.9, 98.7) | 89.8 (69.3, 96.9) | 5 | 17.52 | <0.001 |
| Advanced | 67.4 (58.5, 74.8) | 55.6 (44.1, 65.6) | 62 | 49.48 | |
| Nodal status | | | | | |
| Positive | 70.9 (62.3, 77.8) | 58.9 (47.4, 68.7) | 57 | 48.74 | 0.005 |
| Negative | 95.0 (69.4, 99.2) | 87.6 (58.1, 96.8) | 3 | 11.26 | |
| Surgical therapy | | | | | |
| Yes | 78.7 (70.3, 85.0) | 70.5 (59.6, 78.9) | 37 | 53.07 | <0.001 |
| No | 58.0 (41.8, 71.1) | 42.0 (22.9, 60.1) | 30 | 13.93 | |
| ACT | | | | | |
| Good adherence | 94.4 (79.0, 98.6) | 82.2(55.7, 93.6) | 6 | 15.79 | |
| Poor adherence | 75.6 (65.0, 83.4) | 66.1(52.7, 76.5) | 31 | 35.27 | <0.001 |
| No chemotherapy | 43.7 (27.6, 58.7) | 36.6 (21.1, 52.3) | 30 | 15.95 | |
| Hormonal therapy | | | | | |
| Yes | 94.2 (86.5, 97.5) | 85.2 (72.0, 92.4) | 13 | 35.64 | <0.001 |
| No | 52.6 (40.5, 63.4) | 41.3 (27.6, 54.5) | 54 | 31.36 | |

Abbreviations: ACT: adherence to chemotherapy

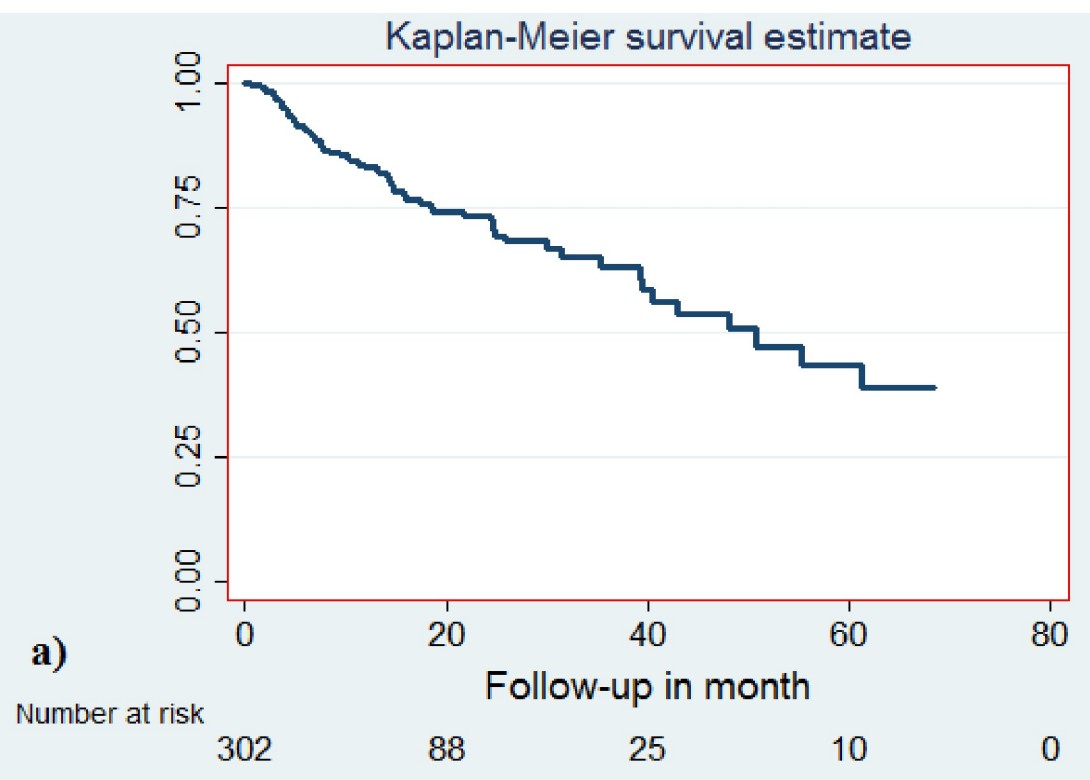

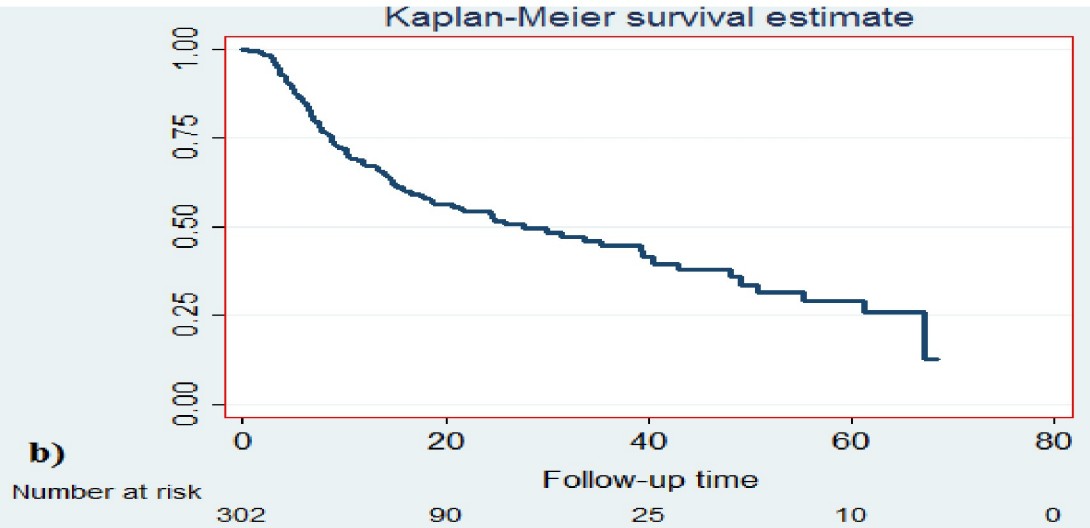

**Fig 2. Kaplan-Meir survival curve showing time to death among female breast cancer patients in South Ethiopia, 2013–2018.** (a) Main analysis (b) Worst-case analysis.

## Discussion

This study estimated the overall survival of BC patients in southern Ethiopia and identified predictors of survival. Our study found that the overall survival rate of patients with BC was at 1 year 83%, at 2 years 73.2%, and at 3 years 63%. Predictors of worse survival included rural residence, time to travel to the hospital >7 hours, duration of symptoms >6 months, and poor adherence to chemotherapy.

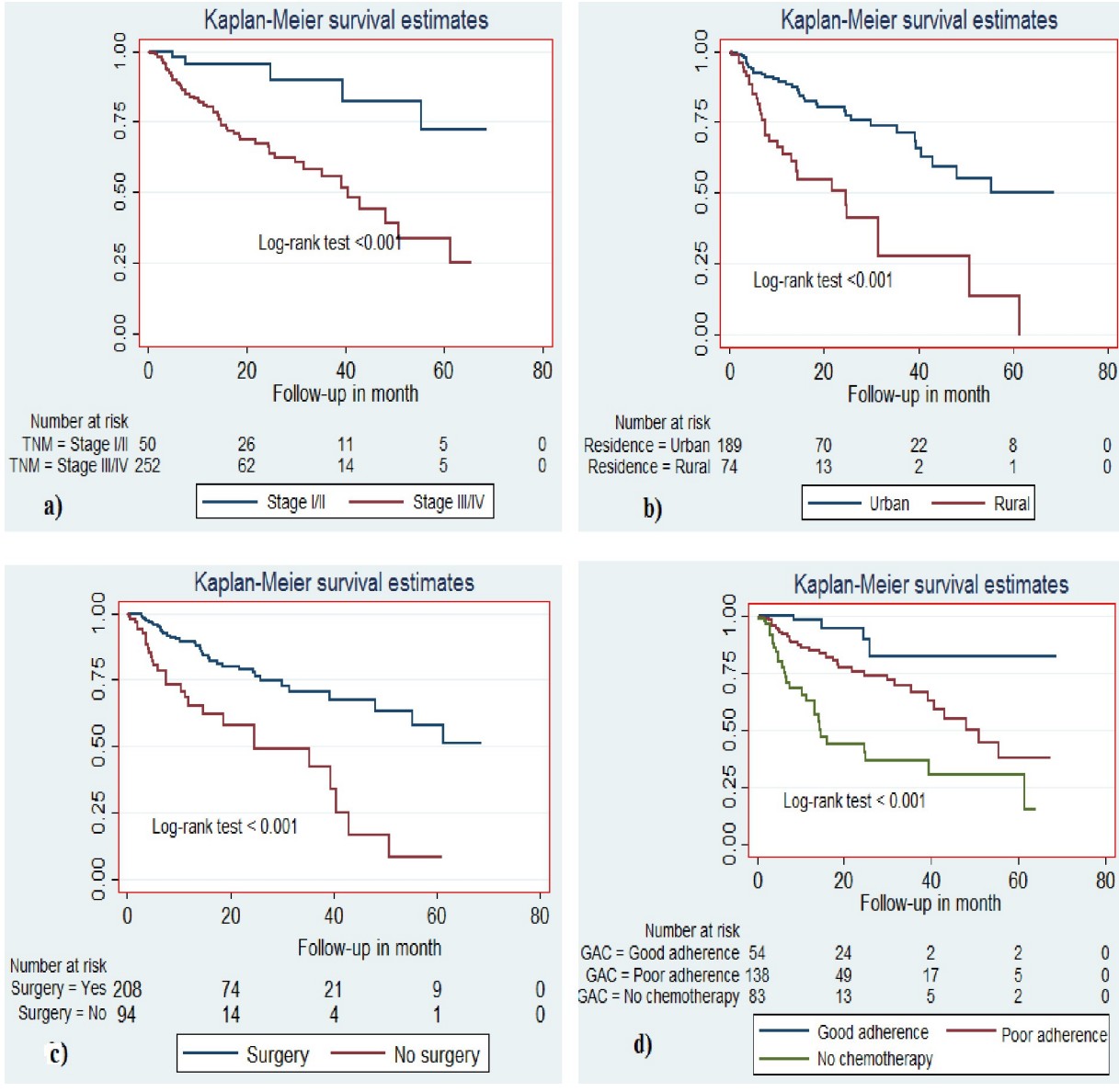

**Fig 3. Kaplan-Meier survival curve among female breast cancer patients in South Ethiopia, 2013–2018.** (a) for the stage of breast cancer (b) for residence (c) for surgical therapy (d) for adherence to chemotherapy.

The two-year survival of BC patients we observed (73.2%) was higher than that observed in other Ethiopian studies. Similarly, a retrospective cohort study conducted in northwest Ethiopia reported a two-year survival rate of 54.2% among BC patients, while a study from western Ethiopia including patients without systemic treatment reported a two-year survival rate of 53% [11]. The two-year overall survival rate in this study (73.2%) is higher than that of a systematic review performed in Iran, which was 67.6% [18]. The difference between these studies could be due to a higher number of losses to follow-up in this study (18%), which in turn affected the actual overall survival. Another reason could be the availability of systemic therapy, as patients have access to it, unlike the western Ethiopia study that reported survival without it. Furthermore, the centre has been providing breast care in southern Ethiopia for many years and is now preparing to offer radiotherapy.

**Table 4. Multivariable Cox regression analysis model for survival of female breast cancer patients in southern Ethiopia, 2013–2018 (n = 302).**

| Covariates | Category | Best-case Scenario | | |
| --- | --- | --- | --- | --- |
| | | CHR (95% CI) | AHR (95% CI) | P-value |
| Patients residence | Urban | 1 | 1 | |
| | Rural | 3.29 (2.00, 5.42) | 2.71 (1.44, 5.09) | 0.002 |
| Travel time to hospital | < 3 hours | 1 | 1 | |
| | 3–6 hours | 1.40 (0.73, 2.66) | 1.11 (0.52, 2.38) | 0.773 |
| | ≥ 7 hours | 2.22 (0.94, 5.20) | 3.42 (1.05, 11.10) | 0.041 |
| Duration of symptoms | 0–6 months | 1 | 1 | |
| | 7–23 months | 1.50 (0.84, 2.67) | 2.63 (1.22, 5.64) | 0.013 |
| | > 23 months | 1.71 (0.87, 3.39) | 2.37 (1.00, 5.59) | 0.048 |
| Stage at diagnosis | Early | 1 | 1 | |
| | Advanced | 4.67 (1.85, 11.76) | 3.01 (1.05, 8.59) | 0.040 |
| Adherence to chemotherapy | Good adherence | 1 | 1 | |
| | Poor adherence | 3.37 (1.19, 9.54) | 2.87 (0.96, 8.58) | 0.032 |
| | No chemotherapy | 8.66 (3.03, 24.73) | 6.69 (2.20, 20.30) | 0.001 |

Abbreviations: AHR: adjusted hazard ratio, CHR: crude hazard ratio, CI: confidence interval.

In contrast to our study, better survival has been observed in South Africa and Iran. Survival after two years is 80% in Soweto, South Africa [19] and 86% in Yazd, Iran [20]. This discrepancy could be due to methodological differences, as the study in Soweto relied on existing records and excluded loss to follow-up, which can lead to an underestimation of the number of deaths and an overestimation of survival [19]. Due to the availability of BC care and treatment, 67% of patients with BC had received combined treatment with surgery, chemotherapy, and radiation in the Iranian study [20] compared to only 50.7% in our study who received surgery and chemotherapy, but not radiation therapy.

In most studies performed high-income countries, even the five-year survival rate is better than the two-year survival rate of our study. The five-year survival rate in Idaho (a rural state of the USA) is 89% for women linked to women's health check programs and 83% for those not linked [21]. Increased survival in high-income countries could be due to earlier presentation to cancer treatment, adequate screening services, and quality of care with the completion of treatment. In the USA, a study has shown that mammography screening increases breast cancer early detection, which leads to a shift in BC stage distribution towards earlier stages and ultimately to better survival rates [22].

In our study, most patients with BC presented at an advanced stage (83.4%). This result is consistent with the finding of a systematic review that compared BC patient survival in low to middle-income countries (LMIC) and high-income countries (HIC). Only 20–50% of patients with BC present in early-stages in LMICs. The reason for diagnosis at an advanced stage in LMICs could be due to the very long delay for consultation, access barriers and quality deficiencies in cancer care and treatment, negative perceptions among patients and the community about disease and treatments, beliefs in the effectiveness of alternative and traditional medicine, and a lack of social networks for support [23, 24].

Most studies in the developed world show an association between an advanced clinical stage of BC and delays between presentation and treatment of greater than three months [23]. In high-income countries, 70% of patients with BC are diagnosed in stages I and II, in contrast to our study where the majority had locally advanced disease [23]. We found that patients with BC with advanced disease have a 3.01 times increased risk of death as compared to those with

early stage disease. The finding is similar to study in Mexico, Hawaii, the USA, Nigeria, and Uganda [25–29]. It is therefore evident that earlier presentation or down staging of BC will have a significant impact on survival rates. To ensure that patients benefit from early detection and timely treatment, it is essential to identify the barriers and facilitators in the local context so interventions can be implemented that address them.

In our study, rural residents were more likely to die from BC than their urban counterparts. This is supported by a meta-analysis conducted in HIC of the US and Europe showing that rural dwellers are 5% less likely to survive cancer [30]. Similarly, a study in Utah revealed that rural residents had a 10% increased risk of death [31]. This variation between urban and rural residents could be due to lower healthcare-seeking behaviour in rural residents [32] and low awareness of BC in rural parts of Sub-Saharan Africa [33].

Other reasons for the rural-urban difference in survival can be explained by comparing the percentage of rural vs. urban women who presented with advanced stage disease (87.8% vs. 79.7%); received treatment after 23 months (28.4% vs. 18.6%); and did not receive chemotherapy (48.6% vs. 31.6%). This implies that awareness creation and improving access to care in rural settings would be beneficial to address rural-urban disparities.

BC patients that present to cancer treatment centres within 7–23 months after the onset of breast complaints had a 2.2 times increased risk of death as compared to those who had presented within six months. This finding is supported by a study in Rwanda that explored patient delays of 6 months and above after the onset of symptoms had increased odds of advanced-stage disease when compared with patients who presented within 3 months' time [34].

In our study, we also found patients who travelled more than 7 hours had an increased risk of death compared to those who travelled less than 3 hours to receive cancer treatment. A longer referral service might have resulted in patients being unable to follow up with their treatment, or that the facility is far from their home, leading to transportation challenges. However, this finding had no significant association in any other reviewed study. The awareness of the public and the facility, as shown in West Wolega [14], might have contributed to early diagnosis and in turn reduce the risk of death. Thus, improving early detection and timely treatment of breast cancer overall must identify and address facilitators and barriers in both urban and rural settings.

We observed limitations in our study. Since the study was retrospective, missing data was an issue. Socio-economic variables and other tumour characteristics were not well addressed because of large amounts of missing data. There may have been other competing causes of death that were not taken into account in these mortality rates, but we consider those rare due to the young average age. Reversed causality cannot be excluded when assessing adherence to chemotherapy and survival, as lack of adherence could lead to early death, but progressive disease under chemotherapy may also lead to the reasonable discontinuation of treatment. Furthermore, for some patients, the outcome was unknown and they were considered to be left-censored and alive. This might overestimate the survival time, which is why we performed a sensitivity analysis on the worst cases analysis. Lastly, only 302 patients were treated for over 6 years, which is remarkably low given the hospital's large catchment area of an estimated 15 million people, with approximately 3 million women above the age of 20The reason for this might be the direct referral of patients from other mid-level hospitals in the region to higher diagnostic or cancer treatment center in the capital.

## Conclusion

In southern Ethiopia, overall survival for patients with BC is low after two years. A rural residence, travel time greater than seven hours, prolonged symptom duration greater than six months, advanced stage of disease, and poor adherence to chemotherapy all predicted poor

survival. These factors indicate possibly low awareness about the disease and difficulties in accessing diagnosis and treatment from remote areas. These issues especially need attention when implementing programs to raise community awareness about the disease and strengthen the capacity for early clinical detection, timely diagnosis, and treatment capabilities across the health care system at the primary, secondary, and tertiary levels. Given that the majority of patents in Ethiopia reside in rural areas, timely cancer diagnosis and completion of treatment can only be achieved when access to care from rural areas is also considered.

## Acknowledgments

We would like to acknowledge the cooperation of the hospital staff and medical directors. Our special thanks also goes to all patients and relatives involved in the phone interviews.

## Author Contributions

**Conceptualization:** Abel Shita, Alemayehu Worku Yalew, Sefonias Getachew.

**Data curation:** Abel Shita, Alemayehu Worku Yalew, Sefonias Getachew.

**Formal analysis:** Abel Shita, Alemayehu Worku Yalew, Sefonias Getachew.

**Funding acquisition:** Abel Shita, Alemayehu Worku Yalew, Sefonias Getachew.

**Investigation:** Abel Shita, Alemayehu Worku Yalew, Edom Seife, Tsion Afework, Aragaw Tesfaw, Zenawi Hagos Gufue, Friedemann Rabe, Lesley Taylor, Eva Johanna Kantelhardt, Sefonias Getachew.

**Methodology:** Abel Shita, Alemayehu Worku Yalew, Eva Johanna Kantelhardt, Sefonias Getachew.

**Project administration:** Abel Shita, Edom Seife, Tsion Afework, Aragaw Tesfaw, Zenawi Hagos Gufue, Friedemann Rabe, Lesley Taylor, Sefonias Getachew.

**Resources:** Sefonias Getachew.

**Software:** Abel Shita, Alemayehu Worku Yalew, Sefonias Getachew.

**Supervision:** Edom Seife, Tsion Afework, Aragaw Tesfaw, Zenawi Hagos Gufue, Eva Johanna Kantelhardt, Sefonias Getachew.

**Validation:** Alemayehu Worku Yalew, Edom Seife, Friedemann Rabe, Lesley Taylor, Eva Johanna Kantelhardt, Sefonias Getachew.

**Visualization:** Alemayehu Worku Yalew, Friedemann Rabe, Lesley Taylor, Eva Johanna Kantelhardt, Sefonias Getachew.

**Writing – original draft:** Abel Shita, Sefonias Getachew.

**Writing – review & editing:** Abel Shita, Alemayehu Worku Yalew, Edom Seife, Tsion Afework, Aragaw Tesfaw, Zenawi Hagos Gufue, Friedemann Rabe, Lesley Taylor, Eva Johanna Kantelhardt, Sefonias Getachew.

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
