## [Decision Letter · Decision Letter 0]

7 Sep 2022

PONE-D-22-21553

Survival and Predictors of Breast Cancer Mortality in South Ethiopia: A Retrospective Cohort StudyPLOS ONE

Dear Dr. Getachew,

Thank you for submitting your manuscript to PLOS ONE. After careful consideration, we feel that it has merit but does not fully meet PLOS ONE’s publication criteria as it currently stands. Therefore, we invite you to submit a revised version of the manuscript that addresses the points raised during the review process. Considering the setting of the study (low income country), it provides a valuable insight into predictors of survival and predictors of mortality in breast cancer patients in Ethiopia; however, there are some major points that need to be revised before publication. One of the main limitation of the study is relatively small sample size for a country with more than 100 millions population. Thus, it should be pointed out that the result are representative of special sub-population.

We look forward to receiving your revised manuscript.

Kind regards,

Mohammad-Reza Malekpour

Academic Editor

PLOS ONE

Journal Requirements:

"The study was funded by School of Public Health Addis Ababa University Ethiopia as part of graduate studies program"

Reviewers' comments:

Reviewer's Responses to Questions

**Comments to the Author**

1. Is the manuscript technically sound, and do the data support the conclusions?

Reviewer #1: Yes

Reviewer #2: Yes

2. Has the statistical analysis been performed appropriately and rigorously? 

Reviewer #1: Yes

Reviewer #2: I Don't Know

3. Have the authors made all data underlying the findings in their manuscript fully available?

Reviewer #1: Yes

Reviewer #2: Yes

4. Is the manuscript presented in an intelligible fashion and written in standard English?

Reviewer #1: Yes

Reviewer #2: Yes

5. Review Comments to the Author

Reviewer #1: Overall assessment:

Shita et al. reported the survival and predictors of mortality among breast cancer patients in South Ethiopia in a retrospective cohort study. Although some studies with a similar topic have been performed in Ethiopia or other countries, I think the study is significant to the field and it can be added to the body of studies with the same subject performed in low resource settings to feed meta-analysis papers. In my opinion, the major strength of the study is the questions that are asked from participants, which were not just focused on histological and TNM staging of the tumor, and included the area of residence, treatment adherence, access to facilities, and so forth. These variables can add to the public health value of the study and reveal barriers that will result in a poor outcomes. As another strength, the results have been presented clearly and organized. However, the manuscript should be revised and improved in discussion to better reflect the results of the study, interpret them and provide solutions.

Overall, I think the manuscript needs revision.

Comments:

• Introduction section

The authors mention that only one study about the survival of breast cancer patients exists in Ethiopia. What about the article published by Tiruneh et al. in 2021?

• Introduction section, The last paragraph

Please restate the objective of the study in this paragraph as a scientific-writing norm in literature.

• Result section, Sampling and data collection procedures

337 patients are too low for a country with a 114 million population. Please mention the population in the south of Ethiopia which is covered by this study. Besides, in the case of available literature, this could be a discussion material that how many patients do not go to hospitals for cancer treatment and are not registered anywhere. As it seems, the cases studied here are only the tip of the iceberg, and the patient’s real crude number is much higher. If available, also present data on the completeness of cancer registries in Ethiopia.

• Discussion, second paragraph

The result cannot be due to the availability of treatments because such treatments are more available in Iran. It could be postulated that the authors should consider the worst scenario as the right one. Please expand your discussion on this issue: according to the literature, which of the two scenarios seems more rational? Besides, you should also add the reference of Tiruneh et al. which is lacking on the reference list.

• Discussion Section

The discussion is not engaging. It focuses on comparing the study results with other studies worldwide; however, lacks providing solutions for the mentioned problems in low-resource settings or examples of countries with successful control of the disease burden. Most of the discussion is regarding how the diagnosis of patients in advanced stages is affecting disease prognosis. The idea is discussed in several separate paragraphs, whereas it can be presented in one organized paragraph, ending with available solutions for disease screening or increasing awareness of the disease, which should be compatible with the countries’ HDI and GDP context. Please also focus and expand your discussion on KAP studies and studies on barriers and facilitators of following breast cancer treatment in accordance with your study.

• Please increase the quality of graphs/figures.

Reviewer #2: This paper reported the survival and predictors of mortality among breast cancer patients in South Ethiopia on 302 breast cancer patients diagnosed from 2013 to 2018. As a developing country in Africa, Ethiopia needs more consideration from health authorities to decrease the burden of diseases. Cancer diagnosis and management are costly procedures that need proper infrastructure and specialized personnel. Therefore, broadly available infrastructures may be lacking in countries such as Ethiopia, and considering related studies to investigate the ongoing diseases is critical. Consequently, I think more studies in this setting should be conducted. The current manuscript needs further revisions to make it sound scientifically acceptable. My comments are as follows:

Evidence and examples:

Major issues:

• As this study is a retrospective, findings may be affected by the study setting. Especially for finding the predicting factors.

• The authors mentioned all data are fully available without restriction. Therefore, please upload the data for this submission or provide any alternate options for receiving the data.

• The discussion storyline is unorganized, and readers cannot find a path in the narration of the discussion section. Also, some short paragraphs could be merged with other related sentences.

• The introduction consisted of some information that is not related to the aim and results of this study.

• For the sentence “The median survival was 50.61 months (IQR=18.38-50.80)”, the median is near the upper quartile. I suggest rechecking the results and providing the data supporting this statement.

• Ethical statement: Please include the approval number of this research if it is available.

• The draft had some grammatical, English fluency, and readability issues. Please address these issues with the assistance of an English expert.

• Please mention this research only studied female breast cancer in the abstract section.

Minor issues:

• Abbreviations were not defined in the abstract: WHO, IQR, AHR, CI.

• Reporting the numbers with one decimal is suggested.

• As this study is a descriptive study, it is recommended to add the study period to the aim of the study.

6. PLOS authors have the option to publish the peer review history of their article (what does this mean?). If published, this will include your full peer review and any attached files.

Reviewer #1: No

Reviewer #2: No

---

## [Author Response · Author response to Decision Letter 0]

24 Jan 2023

Response to reviewers

Reviewer's Point by Point Responses to the Questions

1. Introduction section

The authors mention that only one study about the survival of breast cancer patients exists in Ethiopia. What about the article published by Tiruneh et al. in 2021?

 Response: We greatly appreciate the reviewer’s positive comments. We have carefully

revised the studies conducted in Ethiopia regarding the survival of breast cancer patients and we obtained another recent study, accordingly, we have added the article published by Tiruneh et al. in 2021 in the introduction section (See page 6; line number 141-142).

2. Introduction section, the last paragraph: 

Please restate the objective of the study in this paragraph as a scientific-writing norm in literature.

 Response: We greatly appreciate the reviewer’s positive comments. We have carefully

revised the last paragraph section of the introduction of the manuscript and accordingly, we have restated the objective of the study in this paragraph (See page 4; line number 152-153). 

3. Result section, sampling and data collection procedures

337 patients are too low for a country with a 114 million population. Please mention the population in the south of Ethiopia which is covered by this study. Besides, in the case of available literature, this could be a discussion material that how many patients do not go to hospitals for cancer treatment and are not registered anywhere. As it seems, the cases studied here are only the tip of the iceberg, and the patient’s real crude number is much higher. If available, also present data on the completeness of cancer registries in Ethiopia.

 Response: We greatly appreciate the reviewer’s positive comments. We have carefully

revised the sampling and data collection procedures of the study. Since we have no recent census data, what we have included in the study setting is the total population of the Southern region of Ethiopia, from the 2007 national census. We have not got published data that describes the completeness of the cancer registries in Ethiopia. (See page 6; line number 160- and Page 7, line number 161). We have also mentioned in the limitation part of the study. 

4. Discussion, second paragraph

The result cannot be due to the availability of treatments because such treatments are more available in Iran. It could be postulated that the authors should consider the worst scenario as the right one. Please expand your discussion on this issue: according to the literature, which of the two scenarios seems more rational? Besides, you should also add the reference of Tiruneh et al. which is lacking on the reference list.

 Response: We greatly appreciate the reviewer’s positive comments. A sentence indicating the availability of treatment as a reason for the discrepancy is now omitted. Tiruneh et al. (2021) is now added to the discussion. The discussion is extended suggesting the difference with between the studies could be due to a higher number of losses to follow-up in our study (18%), and this which in turn affected the actual overall survival. Another reason could be the availability of systemic therapy, as patients have access to it, unlike the western Ethiopia study that reported survival without it and Hawassa is active since couple of years ago as preparing to offer radiotherapy 

5. Discussion Section

The discussion is not engaging. It focuses on comparing the study results with other studies worldwide; however, lacks providing solutions for the mentioned problems in low-resource settings or examples of countries with successful control of the disease burden. Most of the discussion is regarding how the diagnosis of patients in advanced stages is affecting disease prognosis. The idea is discussed in several separate paragraphs, whereas it can be presented in one organized paragraph, ending with available solutions for disease screening or increasing awareness of the disease, which should be compatible with the countries’ HDI and GDP context. Please also focus and expand your discussion on KAP studies and studies on barriers and facilitators of following breast cancer treatment under your study.

 Response: thanks for your critical look and positive comments. We have significantly improved in each section to make more engaging as we addressing our messages. In most part we improved the section and addressed our summarizing message at the end of the discussions and significantly revised it. 

6. Please increase the quality of graphs/figures.

 Response: We greatly appreciate the reviewer’s positive comments. We have increased the quality of the available figures (See each figure).

Reviewer number 2

This paper reported the survival and predictors of mortality among breast cancer patients in South Ethiopia on 302 breast cancer patients diagnosed from 2013 to 2018. As a developing country in Africa, Ethiopia needs more consideration from health authorities to decrease the burden of diseases. Cancer diagnosis and management are costly procedures that need proper infrastructure and specialized personnel. Therefore, broadly available infrastructures may be lacking in countries such as Ethiopia, and considering related studies to investigate the ongoing diseases is critical. Consequently, I think more studies in this setting should be conducted. The current manuscript needs further revisions to make it sound scientifically acceptable. My comments are as follows:

Evidence and examples

Major issues

1. As this study is a retrospective, findings may be affected by the study setting. Especially for finding the predicting factors.

Response: We greatly appreciate the reviewer’s positive suggestion. Since it is the nature of retrospective studies, we may get missing data which can influence the generalizability of the study. Despite we have included the raised concern in our discussion part; an appropriate design of these studies can provide relevant data for policymakers and the wider scientific community. 

2. The authors mentioned all data are fully available without restriction. Therefore, please upload the data for this submission or provide any alternate options for receiving the data.

 Response: We greatly appreciate the reviewer’s positive suggestion. We have mentioned in the second section as stated “The data underlying the results presented in the study will be available from the main or corresponding author in reasonable request” from the options listed. The discussion storyline is unorganized, and readers cannot find a path in the narration of the discussion section. Also, some short paragraphs could be merged with other related sentences.

 Response: We greatly appreciate the reviewer’s positive comments. We have carefully revised the entire discussion section of the manuscript; by making the flow of ideas as organized, simple, scientifically sound, and as short as possible (See each paragraph of the discussion section).

3. The introduction consisted of some information that is not related to the aim and results of this study.

 Response: We greatly appreciate the reviewer’s positive comments. We have carefully

revised each piece of information written under the introduction section of the manuscript and we have amended it accordingly (See each paragraph of the introduction section).

4. For the sentence “The median survival was 50.61 months (IQR=18.38-50.80)”, the median is near the upper quartile. I suggest rechecking the results and providing the data supporting this statement.

Response: We greatly appreciate the reviewer’s positive comments. We have carefully

checked the findings by doing the data analysis again and we found that the 25th percentile was not-estimable as rightly seen in the Kaplan Meir curve. Accordingly, we have modified the median survival time (See Page 13; line number 295-296). 

5. Ethical statement: Please include the approval number of this research if it is available.

Response: We greatly appreciate the reviewer’s positive comments. We have added the ethical approval number obtained from the research and ethical committee of the Scholl of Public Health, Addis Ababa University (See page 21; line number 450-451). 

6. The draft had some grammatical, English fluency, and readability issues. Please address these issues with the assistance of an English expert.

Response: We greatly appreciate the reviewer’s positive suggestion. We have consulted English language experts and now we have amended the possible grammatical, and readability issues raised by the experts. 

7. Please mention this research only studied female breast cancer in the abstract section.

Response: We greatly appreciate the reviewer’s positive suggestion. We have specified the study population as female breast cancer patients in the abstract section of the manuscript (See page 2; line number 59). 

Minor issues

1. Abbreviations were not defined in the abstract: WHO, IQR, AHR, CI.

Response: We greatly appreciate the reviewer’s positive suggestion. We have defined each possible abbreviation listed in the abstract, but the confidence interval “CI” should not be listed as it is a common statistical abbreviation (See page 2; line number 52). 

2. Reporting the numbers with one decimal is suggested.

Response: We greatly appreciate the reviewer’s positive suggestion. Even though there is no clear recommendation from the journal regarding decimal point use, we have amended the decimal points write-up accordingly except for the P-value and crude and adjusted hazard ratio results (See each decimal described). 

3. As this study is descriptive, it is recommended to add the study period to the aim of the study.

Response: We greatly appreciate the reviewer’s positive suggestion. We have added the study period, which is the data extraction period, to the aim method section of the manuscript (See page 7; line number 166-167).

---

## [Decision Letter · Decision Letter 1]

22 Feb 2023

Survival and Predictors of Breast Cancer Mortality in South Ethiopia: A Retrospective Cohort Study

PONE-D-22-21553R1

Dear Dr. Getachew,

We’re pleased to inform you that your manuscript has been judged scientifically suitable for publication and will be formally accepted for publication once it meets all outstanding technical requirements.

Kind regards,

Mohammad-Reza Malekpour

Academic Editor

PLOS ONE

Additional Editor Comments (optional):

Reviewers' comments:

Reviewer's Responses to Questions

**Comments to the Author**

1. If the authors have adequately addressed your comments raised in a previous round of review and you feel that this manuscript is now acceptable for publication, you may indicate that here to bypass the “Comments to the Author” section, enter your conflict of interest statement in the “Confidential to Editor” section, and submit your "Accept" recommendation.

Reviewer #1: All comments have been addressed

Reviewer #2: All comments have been addressed

2. Is the manuscript technically sound, and do the data support the conclusions?

Reviewer #1: Yes

Reviewer #2: Yes

3. Has the statistical analysis been performed appropriately and rigorously? 

Reviewer #1: Yes

Reviewer #2: I Don't Know

4. Have the authors made all data underlying the findings in their manuscript fully available?

Reviewer #1: Yes

Reviewer #2: Yes

5. Is the manuscript presented in an intelligible fashion and written in standard English?

Reviewer #1: Yes

Reviewer #2: Yes

6. Review Comments to the Author

Reviewer #1: I appreciate authors for the revisions on the manuscript. I think most of my comments have been addressed andthe manuscript is now in better shape.

Reviewer #2: (No Response)

7. PLOS authors have the option to publish the peer review history of their article (what does this mean?). If published, this will include your full peer review and any attached files.

Reviewer #1: No

Reviewer #2: No

---

## [Editor Report · Acceptance letter]

24 Feb 2023

PONE-D-22-21553R1 

Survival and Predictors of Breast Cancer Mortality in South Ethiopia: A Retrospective Cohort Study 

Dear Dr. Getachew:

I'm pleased to inform you that your manuscript has been deemed suitable for publication in PLOS ONE. Congratulations! Your manuscript is now with our production department. 

Kind regards, 

on behalf of

Dr. Mohammad-Reza Malekpour 

Academic Editor

PLOS ONE